# Circle-U-Net: An Efficient Architecture for Semantic Segmentation

**Feng Sun** [1,†]**, Ajith Kumar V** [2,3,†]**, Guanci Yang** [4,*]**, Ansi Zhang** [5] **and Yiyun Zhang** [6]

1 Experimental Teaching Center for Liberal Arts, Zhejiang Normal University, Jinhua 321004, China; sunfeng@zjnu.edu.cn
2 The School of AI, Bangalore 560002, India; inocajith21.5@gmail.com
3 Lotte Data Communication R & D Center India LLP, Chennai 600113, India
4 Key Laboratory of Advanced Manufacturing Technology of Ministry of Education, Guizhou University, Guiyang 550025, China
5 School of Mechanical Engineering, Guizhou University, Guiyang 550025, China; zhangas@gzu.edu.cn
6 Engineering College, Zhejiang Normal University, Jinhua 321004, China; zhangyuni@zjnu.edu.cn
* Correspondence: gcyang@gzu.edu.cn; Tel.: +86-0851-8437-007
† These two authors contributed equally.

**Abstract:** State-of-the-art semantic segmentation methods rely too much on complicated deep networks and thus cannot train efficiently. This paper introduces a novel Circle-U-Net architecture that exceeds the original U-Net on several standards. The proposed model includes circle connect layers, which is the backbone of ResUNet-a architecture. The model possesses a contracting part with residual bottleneck and circle connect layers that capture context and expanding paths, with sampling layers and merging layers for a pixel-wise localization. The results of the experiment show that the proposed Circle-U-Net achieves an improved accuracy of 5.6676%, 2.1587% IoU (Intersection of union, IoU) and can detect 67% classes greater than U-Net, which is better than current results.

**Keywords:** U-Net; deep learning; object segmentation; attention mechanism; convolutional neural network (CNN)





## 1. Introduction

Convolutional neural networks (CNN) have achieved state-of-the-art experiments in many computer vision tasks, such as attribute recognition, object detection, semantic segmentation, labels optimizing and so on [1–4]. Semantic segmentation is of paramount importance in the computer vision field because of its applications in box-supervised and texture features. Mask R-CNN [5] and U-Net [6] are used in semantic segmentation.

Mask R-CNN was developed by adding a fork, which aimed at predicting and generating a high-quality segmentation mask. Bhuiyan et al. [7] used Mask R-CNN to automatically detect and sort IWPs in the North Slope of Alaska. Mask R-CNN is widely applied in the remote sensing domain. Mahmoud et al. [8] proposed an adaptive Mask R-CNN in multi-class objects, detected in a remote sensing domain. Zhao et al. [9] created polygons using segmentation algorithms based on Mask R-CNN. Li et al. [10] proposed HTMask R-CNN, which is based on Mask R-CNN. HTMask R-CNN could adopt the features of single-object segmentation from Mask R-CNN. Mask R-CNN and U-Net [6] are widely applied in object segmentation. However, there are some limitations to U-Net. For example. the mIoU (mean Intersection of union, mIoU) is low and is unable to detect 2, 4 and 8 classes.

In this paper, we propose a new type of U-Net [6], called Circle-U-Net. Circle-U-Net has 101 layers and is inspired by a residual net from deep learning and a circle from geometry. We prove that Circle-U-Net can segment objects better than other networks, due to both its depth and residual layer; the more profound the network model, the more incredible because it can segment. Moreover, only in a deeper network can we say that

additional functionalities, such as residual layers, attention layers, are helpful. The residual layers that form a circular pattern suggest that the recurring patterns vividly help the network understand specific parts. Our contributions can be summarized below.

(1)     We put forward a Circle-U-Net network with a circle connect model, which exceeds the performance of the attention mechanism. Our network improves 0.78 mIoU than adding the attention mechanism to our model. The circle connects model is robust and capable in object segment, which performs better than most state-of-the-art experiments.

(2)     Circle-U-Net cannot detect 2 classes, while some networks cannot see 2, 4 and 8 classes. In other words, Circle-U-Net has high power to detect than other networks.

(3)     We prove that the proposed method has a better performance in comparison with the state-of-the-art networks. Furthermore, we organize the rest of this paper as follows: Section 2 describes related works and Section 3 reviews the proposed Circle-U-Net structure in detail. Experimental results and comparisons are described in Section 4, followed by conclusions in Section 5.

## 2. Related Work

Olaf et al. [5] present U-Net, which consists of a contracting path to capture context and a symmetric expanding path that enables precise localization. U-Net is fast, and the segmentation of a $512 \times 512$ image takes less than a second on a recent GPU. Bhakti et al. [11] present Eff-UNet, which combines the effectiveness of compound scaled Efficient Nets and the encoder for feature extraction with U-Net decoder for reconstructing the fine-grained segmentation map. The Eff-UNet combines high-level feature information, as well as low-level spatial valuable information, for precise segmentation. Nazanin et al. [12] propose a Squeeze U-Net inspired version of U-Net for image segmentation. The Squeeze U-Net is efficient in both low MACs and memory use. Edgar et al. [13] propose that U-Net-based architecture provides detailed per-pixel feedback to the generator, while maintaining the global coherence of synthesized images by providing global image feedback. Eisuke et al. [14] propose Feedback U-Net using Convolutional LSTM, the segmentation method using the Convolutional LSTM and feedback process. Wei et al. [15] introduce a novel recurrent U-Net architecture that preserves the compactness of the original U-Net. Benjamin et al. [16] propose an enhanced Rotation-Equivariant U-Net for Nuclear Segmentation. RUNet [17] can learn the relationship between a set of degraded low-resolution images and their corresponding original high-resolution images. Reza et al. [18] put forward a bi-directional Co-nvLSTM (BConvLSTM) U-Net with densely connected convolutions for medical image segmentation, taking full advantage of U-Net, bi-directional ConvLSTM, and the mechanism of dense convolutions. Additionally, BConvLSTM combines the feature maps extracted from the corresponding encoding path and the previous decoding up-convolutional layer in a non-linear way. W-Net [19] is a reinforced U-Net for density map estimation. Retina U-Net [20] is the one-stage detector, which combines detection with an auxiliary segmentation. The adaptive triple [21] U-Net with a test-time augmentation can segment common and internal carotid arteries more efficiently and accurately. R2U-Net [22] realizes the power of Residual Network, U-Net, as well as RCNN, which are several advantages for segmentation tasks. Ozan et al. [23] propose Attention U-Net, which is assessed on CT datasets and is for multi-class image segmentation. UNet++ [24] is a nested U-Net architecture which gets an average IoU gain of 3.9 points over U-Net.

## 3. Method

### 3.1. Circle-U-Net

There is ten-block which starts with a $7 \times 7$ block followed by a pooling layer called C1, then followed by C2 with three bottleneck block, C3 with four bottleneck block with one circle connect, C4 with 23 bottleneck blocks with two circles connects, and C5 with three bottleneck blocks and one circle connect. C1 to C5 forms the contracting layers that capture the context of the image, and C6 to C10 forms the expanding layers to achieve

pixel-wise localization and finally forming a segmented image. C6 comprises a merging layer from C4, C7 comprises a merging layer from C3, C8 comprises a merging layer from C2, C9 gets from C1. We have C10 forming the final expanding layer, which produces the output by pixel-wise classification. Figure 1 denotes the Circle-U-Net architecture as described above.

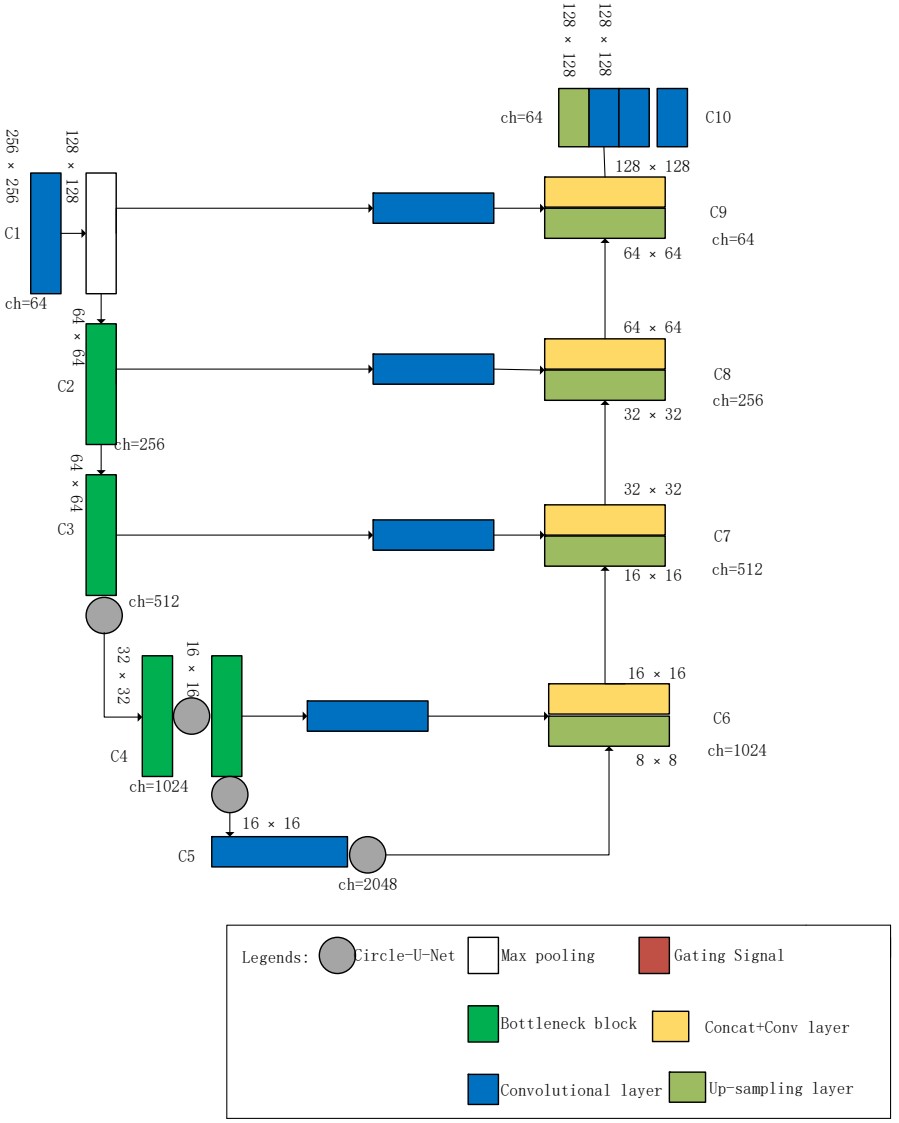

**Figure 1.** Circle-U-Net structure.

In Figure 2, the Concat-Conv Structure suggests that an up-convolution layer is merged with a layer from the contracting path, which is done to add high-resolution features from the contracting path to the expanding path for better localization, and convolution blocks are added successfully to learn to produce precise output.

This is the schematic of our framework, which is described in Figure 3. An input image of 256 × 256 size is pre-processed by resizing, normalization, and then parsing labels. It is then fed to CircleUnet for training, where the contracting path extracts the features and expands the way it localizes. In image segmentation, pixel-wise classification takes place, and we use the CCE loss function. After training with the trained model, we can predict the output.

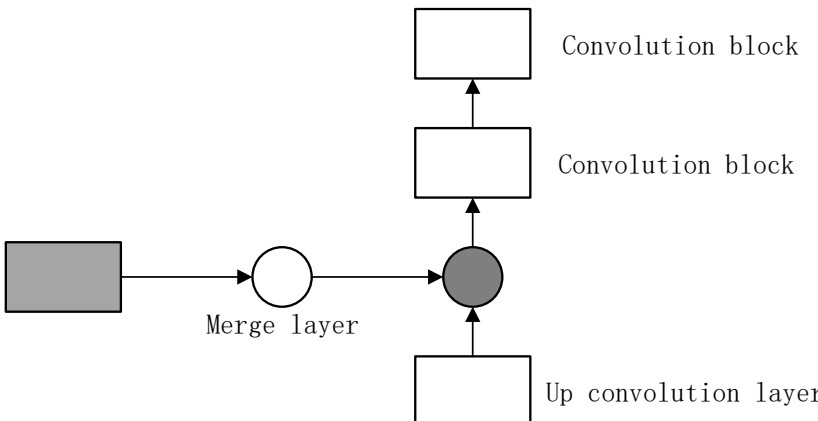

**Figure 2.** Concat-Conv structure.

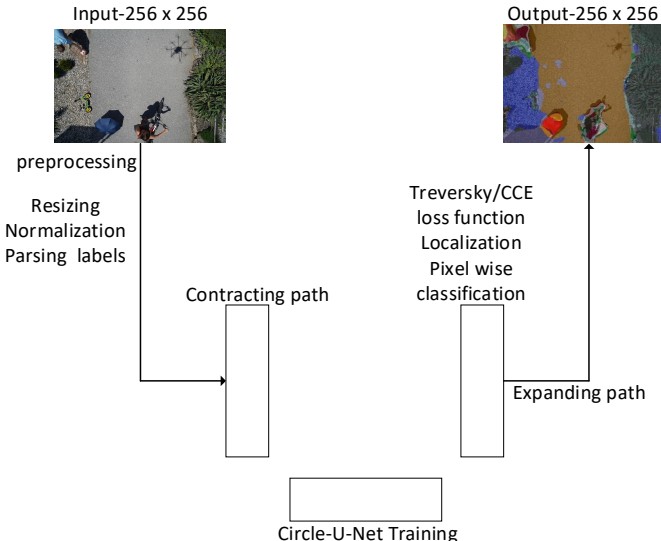

**Figure 3.** The schematic of our framework.

### 3.2. Circle-U-Net with Attention

Figure 4 describes Circle-U-Net with attention network. There are ten blocks, starting with a $7 \times 7$ block, followed by a pooling layer called C1, then followed by C2 with three bottleneck block, C3 with four bottleneck block with one circle connect, C4 with 23 bottleneck blocks with two circle connects and C5 with three bottleneck blocks, and one circle connect. C1 to C5 forms the contracting layers, and C6 to C10 forms the expanding layers to achieve a segmented image. C6 comprises gated Attention with a merging layer from C4, C7 comprises gated Attention with a merging layer from C3, C8 comprises gated Attention with a merging layer from C2, C9 gets from C1.We have C10 forming the final expanding layer, which produces the output by pixel-wise classification.

Bottleneck layers. Residual bottleneck layers are used in the neural network to encourage compressed feature representations. They are leading to the reduction in the number of parameters and matrix multiplication with an increased depth. Residual blocks [25] act as a critical ingredient by acting as a backbone for Circle-U-Net. Figure 5 describes a bottleneck layer that has 3-Convolutional blocks. Table 1 gives the structure of each block in the contracting block.

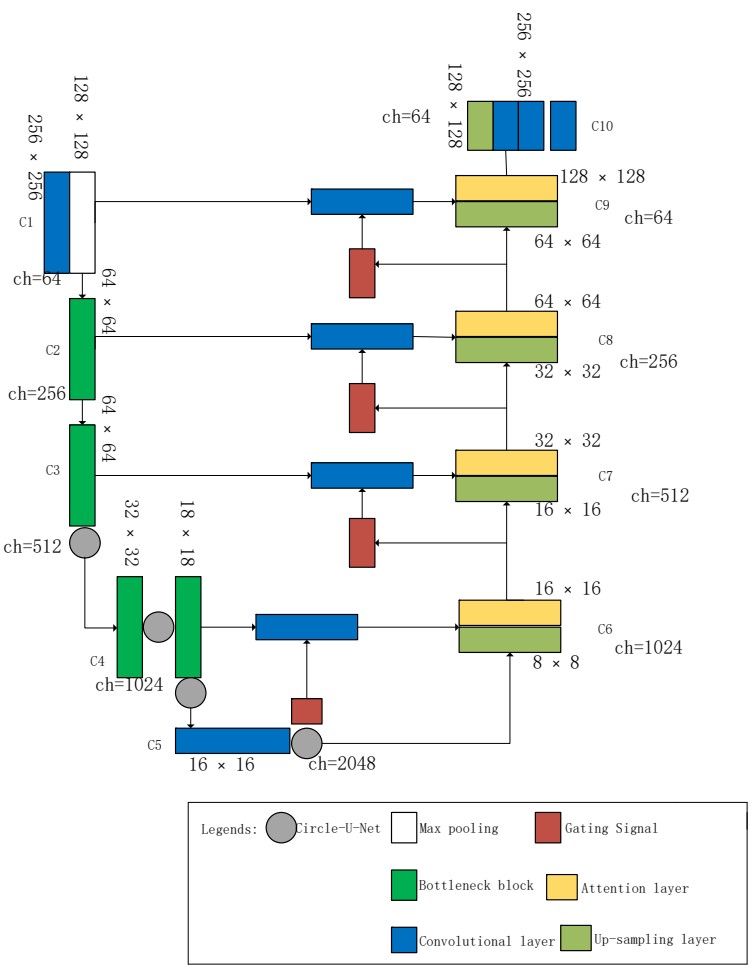

**Figure 4.** Circle-U-Net with attention structure.

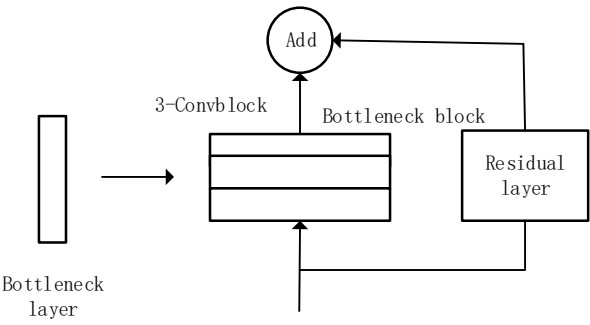

**Figure 5.** Bottleneck Block Structure.

**Table 1.** The Structure of every block.

| Block Name | Number of Bottleneck Layers | Structure |
|:---:|:---:|:---:|
| C2 | 3 | Contains $7 \times 7$ with max-pooling at the first layer |
| C3 | 4 | Bottleneck with residual at the first layer |
| C4 | 23 | Bottleneck with residual at the first layer and circle connect at last |
| C5 | 3 | Bottleneck with residual at first the layer and circle connect at last |

Circle Connect. Circle connect is the new concept we propose through this paper. We take $C_{x-1}$ first layer (i.e., a residual bottleneck block) and connect with the end of Layer x as described in Figure 6. By Table 1 we have proved that using such types of connecting layers for every periodic layer has increased accuracy. When a layer has the information of its far previous it can associate and learn current features with the farther ones. We have four circles connect layers in our architecture, as described in Table 2.

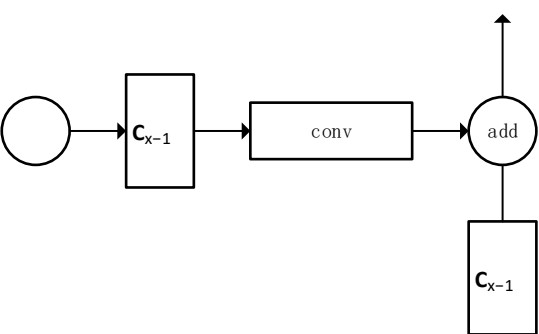

**Figure 6.** Circle connect Structure.

**Table 2.** Circle connect name and layers.

| Circle Connect Name | Connected Layers |
|---|---|
| cc1 | Conv2_1 with conv 3_4 |
| cc2 | Conv 3_1 with conv 4_11 |
| cc3 | Conv 4_11 with conv 4_23 |
| cc4 | Conv 4_1 with conv 5_3 |

Gated signals. The gated signals include resizing the layer feature map to the up-layer feature map, using $1 \times 1$ convolution. The gating signal used with skip connection aggregates image features from multiple image scales and is fed for attention blocks.

Attention gates. Attention gates (AG) is proposed by Schlemper et al. [26]. AG could learn to automatically focus on the sizes and shapes of target structures. AGs are used when the model implicitly learns to suppress irrelevant features and highlights salient features. When AG is used for CNN models such as U-Net, it could increase prediction accuracy and sensitivity. We are using attention gates in the expansion path at C6, C7, C8, and C9 layers in Circle-U-Net with Gated Signal Attention (GSA) network. AG self-learns to focus on the target structure of different shapes and sizes. Where AGs are used, the model implicitly learns to suppress irrelevant features and highlights salient features. Figure 7 describes attention gates, where gating signal and layer x are concatenated and passed on with Relu activation, which when further resampled, produces attention layer.

$$q_{att,i}^l = \varphi^T \left( \sigma_1 \left( W_x^T x_i^l + W_g^T g + b_{xg} \right) \right) + b_\varphi \tag{1}$$

$$\alpha^l = \sigma_2 (q_{att}^l \left( x^l, g; \varnothing_{att} \right) \tag{2}$$

where $\sigma_1(x) \rightarrow$ element-wise non-linearity, $\sigma_2(x) \rightarrow$ normalization function; $x_i^l \rightarrow$ Activation map of a chosen layer i; g -> global feature vector provides information to AGs to point out irrelevant features in $x_i^l$; $W_x, W_g \rightarrow$ Linear transformations; $b_{xg}, b_\varphi \rightarrow$ Bias terms.

Loss function. We have tested our model with categorical cross-entropy loss function and Focal Tversky loss function, and it works better with both the type of loss functions.

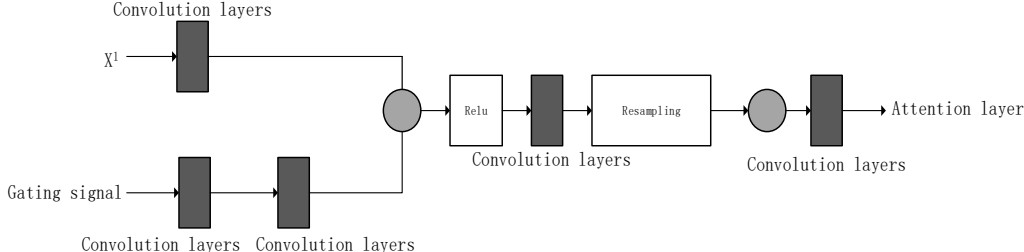

**Figure 7.** Attention gates structure.

Categorical cross-function is described as below.

$$Loss = -\frac{1}{N}\sum_{i=1}^{N}\sum_{c=1}^{C}1_{y_i \in C_{y_i}}\log P_{model}[y_i \in C_e] \tag{3}$$

$N$ is the total number of observations, $i$ denotes specific observation, $C$ denotes the total number of categories, and c denotes a specific category. $\sum_{i=1}^{N}\sum_{c=1}^{C} \rightarrow$ Observations of '$I$', till $N$ and categories '$c$' till '$C$'; $1_{y_i \in C_{y_i}} \rightarrow i$th observation for $c$th category; $P_{model}[y_i \in C_c] \rightarrow$ probability predicted by the model for $i^{th}$ observation which belongs to $c$th category

Focal Tversky loss function is described as below.

$$TI_c = \frac{\sum_{i=1}^{N} p_{ic}g_{ic} + \in}{\sum_{i=1}^{N} p_{ic}g_{ic} + \alpha\sum_{i=1}^{N} p_{\overline{ic}}g_{ic} + \beta\sum_{i=1}^{N} p_{ic}g_{\overline{ic}} + \in} \tag{4}$$

where $TI$ is the Tversky index, $\alpha + \beta = 1$; $p_{ic} \rightarrow$ probability that the pixel $i$ is of the lesion class $c$; $p_{\overline{ic}} \rightarrow$ probability that the pixel $i$ is of the non-lesion class $c$; $g_{ic} \rightarrow$ ground truth that the pixel $i$ is of the lesion class $c$; $g_{\overline{ic}} \rightarrow$ ground truth that the pixel $i$ is of the non-lesion class $c$;

$$FTL_c = \sum_c (1 - TI_c)^{\frac{1}{\gamma}} \tag{5}$$

where $\gamma$ controls the non-linearity loss. We use the focal Tversky loss function. It solves class imbalance as FTL is robust to class imbalances.

## 4. Experiments

### 4.1. Datasets

ICG Drone dataset contains 400 images with 20 label classes prepared for semantic understanding of urban scenes for increasing the safety of autonomous drone flight and landing procedures. The dataset is acquired at an altitude of 5 to 30 m above ground, and imagery shows more than 20 houses in a bird's eye view. This dataset is similar to biomedical segmentation, where there would be the top view of cells or bacteria. Still, it is very different from datasets, such as CamVid or Cityscape, where the datasets would be in a frontal or side view. We split the train and test set randomly to maintain uniformity in class distribution while training. We show the spread of data in the ICG Drone dataset in Figure 8, and we can see that paved areas (37.7%) and grass (19.8%) is occupying most of the dataset.

## ICG Drone Dataset for Semantic Segmentation

| ■ unlabeled | ■ paved-area | ■ dirt | ■ grass | ■ gravel | ■ water |
| ■ rocks | ■ pool | ■ vegetation | ■ roof | ■ wall | ■ window |
| ■ door | ■ fence | ■ fence-pole | ■ person | ■ dog | ■ car |
| ■ bicycle | ■ tree | ■ bald-tree | ■ ar-marker | ■ obstacle | |

**Figure 8.** The constitute of ICG Drone dataset.

### 4.2. Experimental Setup We Use the Machine with RTX 2080 Ti GPU and 256G RAM for Our Experiments

To develop all the models and legally testify them, we use a similar setup. The tensorflow-gpu version is 2.1.4. We train Circle-U-Net for 60 epochs. We set the output height as 256. The batch size is 5. The output width is 256 and the optimizer is Adam for training, and we do not use any other data augmentation techniques. Initially, we faced an overfitting issue when we split the dataset sequentially. Later, when we identified the root cause that the entire dataset is like a video sequence that contains particular objects in one part and inevitable thing in another aspect, we split the train and tested the set randomly. We were able to overcome the overfitting issue. You could visit our website for more commands and usage. Both the dataset and code will be available at https://github.com/sunfeng90/Circle-U-Net, accessed on 20 May 2021.

### 4.3. Comparison to the State of the Art

As shown in Table 3, we can find the sample source and the number of training sample and testing sample.

**Table 3.** Samples Number.

| Source: https://www.tugraz.at/index.php?id=22387 (Accessed on 20 May 2021) | | |
|:---:|:---:|:---:|
| **Total Samples** | **Training Sample** | **Testing Sample** |
| 400 | 360 | 40 |

As shown in Table 4, we can observe from Table 4 that Circle-U-Net (without GSA and Attention) is one of the architectures that outperformed all of the other architectures, irrespective of the loss of function or inaccuracy. The GSA is the abbreviation of Gated Signal Attention. The CCE is the abbreviation of Categorical Cross-Entropy loss. Circle-U-Net (without GSA and Attention) improves 2.1587% IoU and 5.6676% accuracy than U-Net.

**Table 4.** Comparing against the state of the art (unit: %).

| Model | Loss Function | Accuracy | IoU |
|---|---|---|---|
| U-Net [6] | CCE | 69.90883 | 18.462254 |
| Attention U-Net [23] | CCE | 69.0122 | 16.37 |
| Squeeze U-Net [12] | CCE | 72.053033 | 19.439353 |
| ResUNet-a [27] | CCE | 71.906507 | 15.959859 |
| Circle-U-Net (without GSA and Attention) | CCE | 75.576364 | 20.620922 |
| Circle-U-Net (without GSA and Attention) | Trversky | 73.961304 | 19.437675 |
| Circle-U-Net (with Attention) | CCE | 72.75489 | 19.843579 |
| Circle-U-Net (with Attention) | Trversky | 70.40 | 17.076342 |

*4.4. Comparing Top 8 Classes*

U-net is the base architecture, squeeze U-net is developed after Attentation Unet and ResUnet, which is a lightweight architecture. ResUnet (Resnet101) and CircleUnet are heavy architecture. In Attentation Unet, attention layers are used with gating signal on top of Unet base architecture, In Squeeze Unet, squeeze block are used by modifying Unet, In ResUnet we have used residual blocks and in Circle net we have used residual block and circle connect by modifying Unet. In Circlenet101 with GSA we have used residual block, circle connect and attention layers.

Since it is an encoder-decoder type of architecture without any detectors in the first stage, U-Net cannot outperform small objects' criteria. Moreover, as the number of classes increases in many datasets, class imbalance takes place. The imbalance makes the models' accuracy reduce, but a highly available object in every image will be higher. In Table 5, we can find that U-Net and Attention U-Net cannot detect eight classes, whereas Circle-U-Net could not detect only two classes and has considerable accuracy. This experiment proves the Circle-U-Net can segment more objects and achieve considerable mIoU while segmenting.

**Table 5.** Comparing against the mIoU of the state of the art.

| Network Name | Loss Function | Accuracy (Top 8 Classes) | mIoU (8 Classes) | Undetected Classes |
|---|---|---|---|---|
| U-Net [6] | CCE | 55.16 | 0.4132 | 8 |
| Attention U-Net [23] | CCE | 58.26 | 0.4109 | 8 |
| Squeeze U-Net [12] | CCE | 59.84 | 0.4407 | 4 |
| ResUNet-a [27] | CCE | 57.17 | 0.4087 | 2 |
| Circle-U-Net (without GSA) | CCE | 59.29 | 0.4132 | 2 |
| Circle-U-Net (without GSA) | Tversky | 42.07 | 0.2720 | 4 |
| Circle-U-Net | Trversky | 59.30 | 0.4647 | 2 |
| Circle-U-Net | CCE | 56.29 | 0.5891 | 2 |

*4.5. Ablation Study*

To compare the results of Circle-U-Net with attention, and Circle-U-Net without attention. Table 6 tells that Circle-U-Net has performed better than attention architectures.

As shown in Table 4, we can find Circle-U-Net (without Attention) improve 0.78mIoU and 0.82% accuracy than Circle-U-Net (with Attention).

Figure 9 are the predictions of Circle-U-Net and corresponding ground truth.

**Table 6.** Circle-U-Net with or without Attention (unit: %).

| Model | Loss Function | mIoU | Accuracy | Layers |
|---|---|---|---|---|
| Circle-U-Net (without Attention) | CCE | 19.62 | 74.57 | 101+ |
| Circle-U-Net (with Attention) | CCE | 18.84 | 73.75 | 101+ |

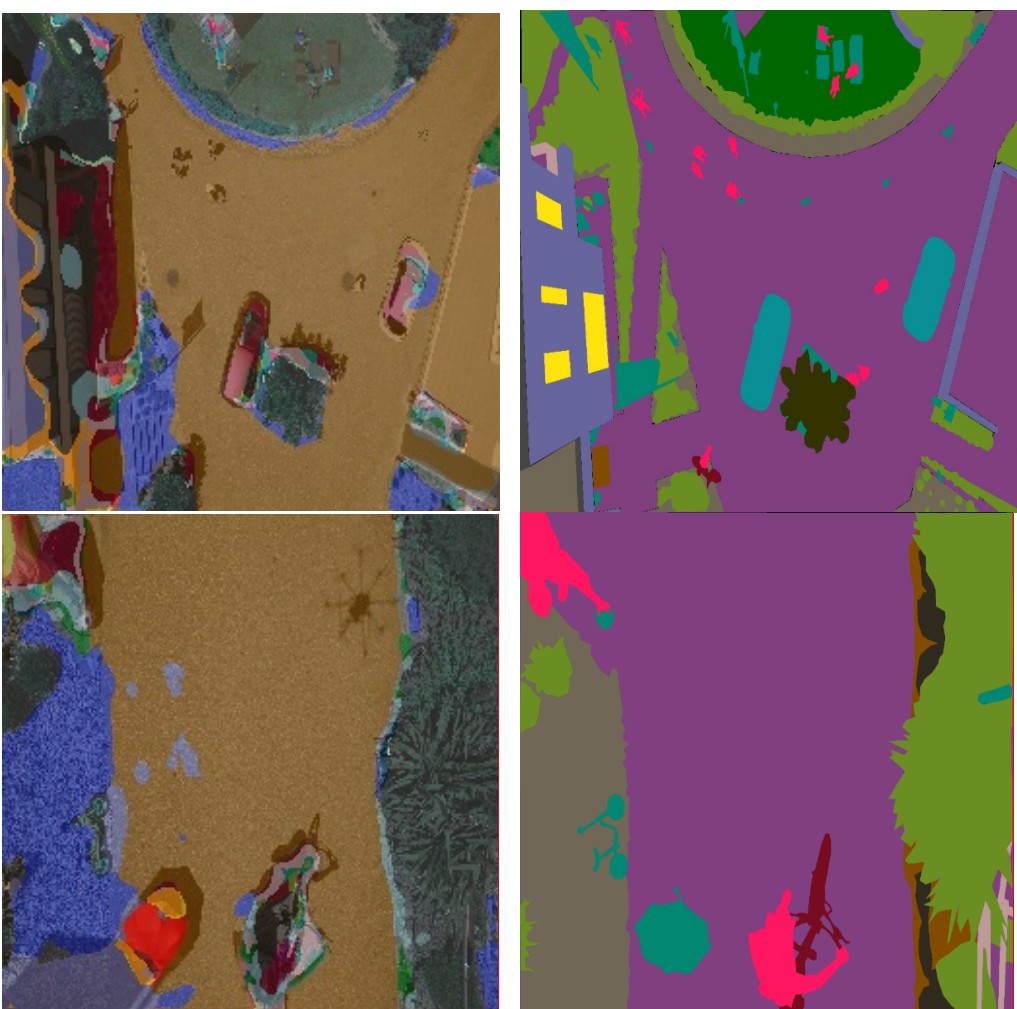

**Figure 9.** The predictions of Circle-U-Net.

## 5. Conclusions

In this paper, we propose a Circle-U-Net which helps to understand the difference between other layers. These deeper layers can recognize more objects and residual connections with circle connect and help to understand the specific part vividly. We can find Circle-U-Net surpasses the attention model through experimental results. In the future, we will apply Circle-U-Net in the services robot domain and put forward a new network about object segmentation.

**Author Contributions:** Conceived of and designed the study, F.S. Implementing the algorithms and analysis of the data, F.S. and A.K.V. Writing, G.Y. Checked the syntax errors of the manuscript and provided the ex-periment machine, A.Z. and Y.Z. All authors have read and agreed to the published version of the manuscript.

**Funding:** This research was funded by the National Natural Science Foundation of China (No.61863005), the Science and Technology Support Plan of Guizhou Province (PTRC[2018]5702, QKHZC[2019]2814,

[2020]4Y056,PTRC[2020]6007, [2021]439), the Experimental Technology and Development Project of Zhejiang Normal University( SJ202123).

**Institutional Review Board Statement:** The study was conducted according to the guidelines of the Declaration of Helsinki, and approved by the Institutional Review Board (or Ethics Committee) of NAME OF INSTITUTE (protocol code XXX and date of approval).

**Informed Consent Statement:** Informed consent was obtained from all subjects involved in the study.

**Data Availability Statement:** Data was obtained from https://github.com/sunfeng90/Circle-U-Net (accessed on 20 May 2021).

**Acknowledgments:** We thank anonymous reviewers for valuable comments that help improve the paper during revision.

**Conflicts of Interest:** The authors declare no conflict of interest.

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
