# Peer review of "Circle-U-Net: An Efficient Architecture for Semantic Segmentation"

_algorithms, doi:10.3390/a14060159_

Round 1
Reviewer 1 Report
This study introduces a novel Circle U-Net architecture for image segmentation. Circle U-Net achieved better performance than the current state of art methods like unet on the ICG drone dataset.
The manuscript is poorly written and hard to follow due to several grammatical errors. The approach looks interesting but is described poorly. The Github link mentioned in the manuscript containing data and code does not exist. The Figure label does not provide sufficient information about the figures.
There are few other comments I want to mention:
- Most of the first 16 references are unnecessary.
- Line 34: No need to create a paragraph for one single sentence.
- Please rephrase lines 38-39. It is not clear what the authors meant to say.
- Please rephrase lines 40-41. It is not clear what the authors intended to say.
- Section 2.1 is unnecessary. Please remove it.
- Related work looks good, but I recommend adding unet variants like nestnet, recurrent unet (r2u-net), and attention unet.
- Add the dimension of each block in figures 1 and 3. This will improve the readability.
- Please clearly explain the difference between a gating signal and an attention gate is.
- Table 1 bottle neck should be written as bottleneck.
- Why are the same footnotes repeated several times?
- Section 4.2: Please explain how the hyperparameters like epochs number have been selected for the proposed approach and the models compared.
- Since the number of samples is low, the author should perform five-fold cross-validation to show the stability of the result.
Author Response
Firstly, I want to thank u for useful comments that help improve the paper during revision.
Secondly, the following are the conditions about these comments one by one:
1) I amended several grammatical errors in the current draft version. I'm so sorry for these errors.
2) I set the GitHub link to be private in the past. Now I set it to be public. U can review it directly.
3) I removed the first 16 references except the first one. The first one described the U-Net which is the based model of
our work. So I didn't remove it. If u don't think so, please tell me, I will remove it in the next draft version.
4) I removed Line 34 which was set to be one single sentence.
5) I rephrased lines 38-39 and 40-41. Please review them again. Thanks a lot.
6) I removed Section 2.1.
7) I added nest net, recurrent unet, and attention unet in related work. Thanks for this suggestion.
8) About the suggestion 'Add the dimension of each block in figures 1 and 3. This will improve the readability. '
I updated them. I'm so sorry for this.
9) About the suggestion 'Please clearly explain the difference between a gating signal and an attention gate is.'.
Okay.
gating signal - resizing down layer feature map to uplayer feature map using 1x1 convolution
attention gate - AGs are used model implicitly learns to suppress irrelevant features and highlights salient features
10)About the suggestion 'Table 1 bottleneck should be written as a bottleneck.'. I fixed bottleneck to bottleneck.
11) About the suggestion 'Why are the same footnotes repeated several times?'. I deleted the same footnotes.
12) About the suggestion 'Section 4.2: Please explain how the hyperparameters like epochs number have been selected for the proposed approach and the models compared. ',
I added some hyperparameters in section 4.2.
13) About the suggestion 'Since the number of samples is low, the author should perform five-fold cross-validation to show the stability of the result.' I don't have enough to do this. Because the editor only gives us 10 days to update the paper. I'm so sorry for this.
Lastly, thank you for reviewing my paper again.

Reviewer 2 Report
I have a couple of concerns with the presented introduction/ methodology that should be addressed. These concerns are a result of a general imbalance between introduction/context information and methodological/analysis information. My major comments and questions are as follows:
- The introduction section is not clear. Recently the state of the arts deep learning technique Resnet based MASK RCNN was established in remote sensing mapping and land cover applications. It would help the manuscript tremendously if the state-of-the-art was more streamlined in the introduction section. Please introduce MASK RCNN based semantic segmentation example using high-resolution satellite imagery application. This would make it much easier for the reader to understand, how this study fits into the research context. What is the uniqueness of the proposed algorithm and its potential impacts, over other recently established states of the art Resnet based Mask RCNN semantic segmentation methods for Remote Sensing application (Bhuiyan, et al.2020; Li, et al. 2021, Mahmoud, et al 2020, Zhao et al 2018, and so on)? They successfully developed an automatic extraction framework for remote sensing applications from high spatial resolution optical images using CNN architecture in a large-scale application. Please introduce these latest advanced research works, and their potential impact, and their limitation in terms of algorithms. Then introduce your proposed methodology and the novelty of your proposed methodology. The authors should explain this aspect in the introduction section.
- Li, Y.; Xu, W.; Chen, H.; Jiang, J.; Li, X. A Novel Framework Based on Mask R-CNN and Histogram Thresholding for Scalable Segmentation of New and Old Rural Buildings. Remote Sens. 2021, 13, 1070.
- Bhuiyan et al. 2020 “Use of Very High Spatial Resolution Commercial Satellite Imagery and Deep Learning to Automatically Map Ice-Wedge Polygons across Tundra Vegetation Types.” J. Imaging 2020, 6, 137.
- Mahmoud, A., et al. "Object Detection Using Adaptive Mask RCNN in Optical Remote Sensing Images." Int. J. Intell. Eng. Syst 13 (2020): 65-76.
- Zhao, Kang, et al. "Building extraction from satellite images using mask R-CNN with building boundary regularization." Proceedings of the IEEE Conference on Computer Vision and Pattern Recognition Workshops. 2018.
- Can you add MASK RCNN algorithms and compared them to your results?
- Can you provide a high impactful schematic diagram to understand the proposed research framework where the big impact of the results can be presented?
- Can you please provide a table for validation/training/testing samples along with other information such as sample number, patch, sample source, training and validation sites, repository, etc?
- How did you utilize the transfer learning strategy? Did you use trained weight? It is unclear what independent evaluation is meant here. You should present the transferability of the Deep Learning Model in your application.
- Can you provide few feature maps during optimization?
- How do the authors come up with the current optimized DL structure? Can you show some results for model optimization? How did you create DL optimized model without showing any fundamental results? For DL optimization you need to optimize your model for different losses such as Smooth-L1 loss; bounding box loss; classifier loss; binary cross-entropy loss; RPN bounding box loss; RPN classifier loss.`
- Can you explain more about overfitting issues?
- In your result section, where are the classification accuracy results in terms of mean Average Precision?
- In the discussion section, you should discuss more your results vs previous research-based semantic segmentation.
- you can see the latest high impactful research to see the Influence of fusion on classification for image application.
Witharana, Chandi, et al. "Understanding the synergies of deep learning and data fusion of multispectral and panchromatic high-resolution commercial satellite imagery for automated ice-wedge polygon detection." ISPRS Journal of Photogrammetry and Remote Sensing 170 (2020): 174-191.
Yang, J.; Zhao, Y.-Q.; Chan, J.C.-W. Hyperspectral and Multispectral Image Fusion via Deep Two-Branches Convolutional Neural Network. Remote Sens. 2018, 10, 800. https://doi.org/10.3390/rs10050800
Author Response
Firstly, I want to thank u for useful comments that help improve the paper during revision.
Secondly, the following are the conditions about these comments one by one:
1) I added four papers that you gave in the past about Mask RCNN and remote sensing into the introduction section. Thanks for your
great suggestions.
2) I explained the four papers in the introduction section. Thanks a lot.
3) About the suggestion "Can you add MASK RCNN algorithms and compared them to your results?". I want to answer "yes, I can.
But the editor gives me ten days to edit the paper. I can't finish these experiments within 10 days. It may spend us 1 month.
I'm so sorry for this. Please forgive me for this.
4)About the suggestion, "Can you provide a high impactful schematic diagram to understand the proposed research framework where the big impact of the results can be presented?"
I cannot solve it. Could u give me an example? thanks a lot.
5) About the suggestion, "Can you please provide a table for validation/training/testing samples along with other information such as sample number, patch, sample source, training, and validation sites, repository, etc?"
I provide table1 in the draft.
6) About the suggestion, "In the discussion section, you should discuss more your results vs previous research-based semantic segmentation.
".
I updated the draft in section4.4.Thanks for this suggestion.
7) About the suggestion, "How did you utilize the transfer learning strategy? Did you use trained weight? It is unclear what independent evaluation is meant here. You should present the transferability of the Deep Learning Model in your application."
my answer is "we trained from scratch".
8) About the suggestion, "Can you provide few feature maps during optimization?".
We need some time for this. We cannot finish within 10 days. I'm sorry for this.
9) About the suggestion, "How do the authors come up with the current optimized DL structure? Can you show some results for model optimization? How did you create DL optimized model without showing any fundamental results? For DL optimization you need to optimize your model for different losses such as Smooth-L1 loss; bounding box loss; classifier loss; binary cross-entropy loss; RPN bounding box loss; RPN classifier loss.`"
We didn't do any optimization for these different losses. I'm sorry for this.
10) About the suggestion, "Can you explain more about overfitting issues?"
we selected train images randomly so we were able to overcome it
11) About the suggestion, "In your result section, where are the classification accuracy results in terms of mean Average Precision?"
I am not sure this but this is the function https://github.com/sunfeng90/Circle-U-Net/blob/09364703232537a57934d9d9b702b326a04db1d1/Utils/drone_metrics.py#L1151

Reviewer 3 Report
Dear authors,
First of all, I would like to congratulate to the authors for this work. Absolutely, it is an interesting topic and a new improvement of U-Net algorithm is described. It is one of the major scientific impact topics at the moment. So, it is a novelty project in this domain.
In general terms, this work is well organized however, the introduction part, related work and experiments sections should be improved. Moreover, English redaction needs a review. It is important to preserve the correct English forms (So, expressions like “can’t” or informal syntaxes forms should be corrected. Now, I will suggest several points that should do:
- Introduction and related work can be more descriptive and more extended version. One of the important points of this work is the comparison the “new improvement” respect to other techniques. In order to understand the proposal of the authors is important to take account the main features that should be studied for the comparison.
- The key of this work is the comparison between the new Circle U-Net respect the traditional U-Net algorithm. Then, more examples on the experiment section is required. In section 4.5, only one test or one example is showed. In my opinion, it is necessary to do more tests or more examples validating the performance and the improvement of Circle U-Net. Moreover, it is important to cite the figures on the text. For example, Fig8 is not cited in this section.
- Following the analysis of this work, in my consideration, it is the essential step to write a discussion section. It is true that the authors tried to do it, however, it is not clear. In fact, it is one of the most interest section for the readers and scientific community.
- English style and grammar.
- Quality of the images can be improved (high resolution)
As a conclusion, I have any doubt that this work has a great impact and interest for research. In order to make value and take advantage of the big effort to implement this technique, it is important to improve the quality of the paper. For this reason, in my concern, it still has some major revisions. All my suggestions and comments are focused on a constructive way in order to improve the quality of your work.
Best regards.
Author Response
Firstly, i want to thank u for useful comments that help improve the paper during revision.
Secondly, following are the conditions about these comments one by one:
1)I amended the grammar errors.
2)I updated the introduction and related part.
3) I want to do more tests or more examples validating the performance. But there are only ten days to update
this paper. The Editor only gave me ten days.
4) I cited the Fig8 in the section.
5) I amended the English style and grammar.
6) About the suggestion, "he key of this work is the comparison between the new Circle U-Net respect the traditional U-Net algorithm. Then, more examples on the experiment section is required. In section 4.5, only one test or one example is showed. In my opinion, it is necessary to do more tests or more examples validating the performance and the improvement of Circle U-Net. Moreover, it is important to cite the figures on the text. For example, Fig8 is not cited in this section.
Following the analysis of this work, in my consideration, it is the essential step to write a discussion section. It is true that the authors tried to do it, however, it is not clear. In fact, it is one of the most interest section for the readers and scientific community."
In section 4.5 values from Table 3 is taken and shown
we tried a 256x256 image
Fig 8 its not able to predict car persons but its able to predict paved area , grass
Lastly, thank you for reviewing my paper again.

Round 2
Reviewer 1 Report
I have no further questions.
Author Response
I want to thank u for useful comments that help improve the paper during revision again.
Reviewer 2 Report
I suggest to authors that you do not have to apologize for anything if you have a logical explanation. If you are not able to address any comments you should explain in a professional way instead of an apology. In the future, I will expect a more mature response while preparing a response document. Overall, all the authors address most of the comments which will significantly improve the paper.
Few minor comments:
- I am providing few literature reviews for the schematic diagrams in response to your reply “4)About the suggestion, "Can you provide a high impactful schematic diagram to understand the proposed research framework where the big impact of the results can be presented?" I cannot solve it. Could u give me an example? thanks a lot.”
See figure 2
Li, et al: A Novel Framework Based on Mask R-CNN and Histogram Thresholding for Scalable Segmentation of New and Old Rural Buildings. Remote Sens. 2021, 13, 1070. https://doi.org/10.3390/rs13061070
or,
See figure 2
Witharana, et al: An Object-Based Approach for Mapping Tundra Ice-Wedge Polygon Troughs from Very High Spatial Resolution Optical Satellite Imagery. Remote Sens. 2021, 13, 558. https://doi.org/10.3390/rs13040558
then explain in detail in terms of your diagram
- You did not address the overfitting issue. You should write more about this issue.
- The caption of figures, table format, and reference styles are mandatory to change according to the journal format
Author Response
Firstly, I want to thank u for useful comments that help improve the paper during revision again.
Secondly, the following are the conditions about these comments one by one:
1) About the suggestion, "Can you provide a high impactful schematic diagram to understand the proposed research framework where the big impact of the results can be presented?"
My answer is "I updated it in the draft. You can read Figure 3".Thanks a lot.
2) About the suggestion"You did not address the overfitting issue. You should write more about this issue."
My answer is "I updated it in the draft in Section 4.2".Thanks a lot.
3) About the suggestion "The caption of figures, table format, and reference styles are mandatory to change according to the journal format''.
My answer is "Thanks a lot. I updated them in the current version which is by consulting the template."
Lastly, thank you for reviewing my paper again.

Reviewer 3 Report
Dear authors,
I would like to congratulate to the authors for this work. All my suggestions and comments were focused on a constructive way in order to improve the quality of this work. It is a pleasure to inform that, after validating your new contributions, your work has been successfully improved. Therefore, the new updated version of this work is ready to be published.
I encourage you to continue working on this topic. My best wishes for all of you.
Best regards,
Author Response

(The authors gave the same response as above.)
